# Hidden Malnutrition in Overweight and Obese Individuals with Chronic Heart Failure: Insights from the Pro-HEART Trial

**DOI:** 10.3390/nu17162694

**Published:** 2025-08-20

**Authors:** Angelina P. Nguyen, Jennifer Kawi, Rebecca Meraz, Kelly L. Wierenga, Alona D. Angosta, Michele A. Hamilton, Gregg C. Fonarow, Lorraine S. Evangelista

**Affiliations:** 1Louise Herrington School of Nursing, Baylor University, Dallas, TX 75246, USA; angelina_nguyen@baylor.edu (A.P.N.); rebecca_meraz@baylor.edu (R.M.); kelly_wierenga@baylor.edu (K.L.W.); alona_angosta@baylor.edu (A.D.A.); 2Cizik School of Nursing, The University of Texas Health Science Center at Houston, Houston, TX 77030, USA; jennifer.kawi@uth.tmc.edu; 3Smidt Cedars-Sinai Heart Institute, Los Angeles, CA 90048, USA; michele.hamilton@cshs.org; 4University of California Los Angeles Health, Los Angeles, CA 90095, USA; gfonarow@mednet.ucla.edu; 5Sue & Bill Gross School of Nursing, University of California Irvine, Irvine, CA 92697, USA

**Keywords:** heart failure, malnutrition, obesity paradox, nutritional deficiencies, overweight and obese individuals

## Abstract

**Background/Objectives:** Poor nutritional status and cachexia have been well-documented as predictors of adverse outcomes in individuals with chronic heart failure (HF). However, despite obesity being a common observation in this patient population, a growing body of evidence indicates that these individuals may still suffer from nutrient deficiencies and malnutrition. This study aimed to characterize the food and nutrient intake of participants enrolled in the Pro-HEART clinical trial—a study evaluating dietary interventions in overweight and obese individuals with HF—and to compare their consumption patterns to national nutritional guidelines. **Methods:** A cohort of 92 overweight and obese individuals with chronic HF enrolled in the Pro-HEART trial. Baseline food consumption was assessed via a validated 3-day Food Record. The data were analyzed using descriptive statistics to evaluate adherence to recommended intake levels for energy, macronutrients (fat, carbohydrates, protein), and key micronutrients. **Results:** Among the 92 participants, 41% exceeded fat intake recommendations, 73% surpassed guidelines for saturated fat, and 95% consumed excessive sodium. Despite adequate caloric intake, many individuals failed to meet recommended levels for key micronutrients known to influence inflammation and metabolic regulation, including vitamin D, calcium, magnesium, and potassium. **Conclusions:** These findings suggest that overweight and obese individuals with HF, despite their excess body weight, exhibit dietary patterns that place them at risk of malnutrition. The results underscore the necessity of nutritional assessments and interventions in this population to address deficiencies that may contribute to the metabolic and inflammatory abnormalities associated with HF.

## 1. Introduction

Heart failure (HF) is a chronic, progressive condition that impacts millions globally. It occurs when the heart is unable to circulate sufficient blood to satisfy the body’s requirements; it is a primary contributor to cardiovascular disease and mortality, complicating long-term management [1]. A multidisciplinary approach is crucial for enhancing patient outcomes [2]. Although drug treatments and device-based therapies remain the primary methods of treating diseases, an increasing number of individuals are recognizing the significance of nutrition in determining the progression of diseases, the severity of symptoms, and their overall quality of life [3,4]. Poor prognoses have been associated with malnutrition, whether it manifests as cachexia or more subtle micronutrient deficiency. Nevertheless, the management of HF often fails to prioritize dietary evaluations [3,5,6,7].

In the past, nutritional concerns in HF have primarily been addressed in the context of underweight individuals with cachexia, a condition in which they experience a significant loss of muscle mass and inadequate food intake, which exacerbates their health [8]. However, an overlooked population in HF research consists of overweight and obese individuals, who, despite excess adiposity, frequently exhibit nutrient deficiencies and imbalanced dietary habits [9]. This contradiction, referred to as the “obesity paradox,” contradicts the prevalent notion that being overweight equates to adequate food intake [10].

Recent evidence indicates that even patients with elevated body mass index (BMI) can exhibit true cachexia or sarcopenic obesity—characterized by disproportionate loss of skeletal muscle mass despite excess adipose tissue [8,11]. Because the BMI conflates lean and fat mass, relying solely on weight may overlook profound muscle wasting and metabolic dysfunction. Studies using body composition tools (e.g., dual-energy X-ray absorptiometry [DEXA], bioelectrical impedance) and functional measures (handgrip strength, gait speed) reveal that up to 20–30% of overweight or obese patients with HF meet cachexia criteria, with worse exercise capacity and higher mortality than their non-cachectic peers. Recognizing hidden cachexia alongside nutrient deficiencies underscores the need for comprehensive nutritional and functional assessment in all patients with HF, regardless of body size.

Data also suggest that a significant number of overweight and obese individuals with HF ingest meals that are calorically dense but nutritionally deficient. These diets do not meet the requisite levels of essential macronutrients and micronutrients, and they exceed the thresholds for detrimental fats, sodium, and processed sugars [12]. These anomalies can exacerbate metabolic inefficiency, inflammation, and muscular atrophy, all of which are detrimental to HF [13]. Excessive consumption of fat, particularly saturated fat, and sodium, has been linked to aggravating HF pathogenesis through inflammation and neurohormonal dysfunction [14]. Inadequate protein and critical mineral consumption, such as calcium, vitamin D, and zinc, can cause muscle atrophy, reduced immunological function, and disease progression [15]. A recent study has focused on the dietary habits of individuals with HF; however, the nutritional condition of those who are overweight or obese is mostly unknown [13]. It is imperative to address this knowledge gap, as targeted therapies may be cost-effective, non-pharmacological methods of enhancing patient outcomes in conjunction with standard HF care [5].

The purpose of this study is to take a comprehensive look at the diets of overweight and obese individuals with HF who are participating in the Pro-HEART trial [16]. This clinical trial compares high-protein diets versus standard-protein diets in this population. The specific aims of this sub-study were to determine which nutrients are excessively high or low in the diets of overweight and obese individuals with HF by comparing them to national recommendations. This will enable us to better understand the dietary concerns associated with this unique population. These findings could be used to develop tailored nutritional regimens aimed at improving dietary guidance, maintaining muscle mass, and improving long-term health outcomes for persons with HF [17].

While malnutrition in obesity has been documented in general populations, comparatively little is known about the detailed dietary patterns and micronutrient profiles of overweight and obese individuals living with chronic HF. To our knowledge, this study is the first to leverage baseline data from a rigorously controlled clinical trial (Pro-HEART) to (1) quantify macro- and micronutrient intakes using a validated three-day food record and the Nutrition Data System for Research; (2) benchmark those intakes specifically against the 2020–2025 Dietary Guidelines for Americans within a chronic HF cohort; and (3) illuminate precise nutritional excesses and deficits (e.g., sodium, saturated fat, vitamin D, calcium, potassium) that may exacerbate HF pathophysiology despite elevated BMI. These novel insights provide a critical foundation for developing HF-tailored nutritional interventions that go beyond weight reduction to optimize metabolic and functional outcomes in this vulnerable population.

## 2. Methods

### 2.1. Study Design and Participants

Pro-HEART is a multi-center, open-label, randomized clinical trial conducted at two University of California health centers (Los Angeles and Irvine) between August 2009 and May 2013, that was designed to evaluate the impact of dietary macronutrient composition on cardiometabolic risk and functional status in overweight and obese patients with chronic HF, verified as a prior hospitalization for HF with objective evidence of left ventricular ejection fraction ≤40% and/or natriuretic peptide elevation (BNP > 100 pg/mL or NT-proBNP > 300 pg/mL) according to criteria from the American Heart Association (AHA), the American College of Cardiology (ACC), and the Heart Failure Society of America (HFSA) [18]. Overweight and obesity were classified by BMI of 25.0–29.9 kg/m^2^ and ≥30.0 kg/m^2^, respectively. Metabolic abnormalities were defined per NCEP ATP III as the presence of at least two of the following: fasting glucose ≥100 mg/dL, triglycerides ≥ 150 mg/dL, HDL-cholesterol < 40 mg/dL (men) or <50 mg/dL (women), blood pressure ≥ 130/85 mmHg, or waist circumference > 102 cm (men) or >88 cm (women). Of the 61 individuals who consented, 52 (85%) met the eligibility criteria and were randomized equally to either a high-protein or standard-protein diet. Key inclusion criteria were age 18–75 years, BMI ≥ 25 kg/m^2^, stable guideline-directed HF therapy for at least 30 days, and either type 2 diabetes mellitus or at least two components of metabolic syndrome. Exclusion criteria included end-stage renal or hepatic disease, recent weight change >5 kg in the prior 3 months, active malignancy, or inability to comply with dietary intervention [16].

This study is listed on ClinicalTrials.gov (NCT1423266) and was approved by the Institutional Review Board at the University of California, Irvine (HS#2010-7972). The trial followed the CONSORT guidelines for pilot and feasibility randomized controlled trials, ensuring that the techniques were sound and the reporting was clear [19].

### 2.2. Dietary Intervention Tailoring

In addition to the primary Pro-HEART diet plan, individuals with diabetes, ischemic heart disease, or chronic kidney disease received tailored recommendations following current clinical practice guidelines. Individuals diagnosed with ischemic heart disease were advised to adhere to a Mediterranean diet. Individuals should consume less than 7% of their daily caloric intake from saturated fat, increase their intake of monounsaturated fat (mostly sourced from olive oil and nuts), and include oily fish (such as salmon and mackerel) in their diet at least twice weekly. Sodium consumption was restricted to under 2300 mg per day to help regulate blood pressure.

Individuals with chronic kidney disease (eGFR < 60 mL/min/1.73 m^2^) or confirmed proteinuria had their daily protein intake restricted to 0.8 g/kg of ideal body weight, with an emphasis on high-biological-value proteins. Participants were instructed to distribute their protein consumption evenly over the day to reduce glomerular hyperfiltration. At the commencement of the trial and again at week 12, 24 h urine urea nitrogen levels were assessed to see if the protein targets were achieved.

Weekly individual counseling sessions reinforced all individualized recommendations. We collaborated with each participant’s cardiologist or nephrologist to ensure the safe integration of dietary and pharmacologic therapies. For instance, we modified the diuretic dosage or initiated sodium-glucose cotransporter 2 (SGLT2) inhibitors.

### 2.3. Dietary Assessment

Dietary intake for each participant was assessed using a three-day weighed food record, comprised of two non-consecutive weekdays and one weekend day [6]. Participants were supplied with a digital kitchen scale (accurate to ±1 g), a standardized set of measuring cups and spoons, and a photographic portion-size guide depicting commonly consumed foods. They were instructed to weigh and record every food and beverage consumed, along with the preparation method, brand name, and condiments, immediately at the time of intake. Within one week of completing their records, each participant met face-to-face with a registered dietitian, who systematically reviewed the entries, probed for any omitted items (for example, snacks or dressings), clarified ambiguous descriptions (such as “one bowl” versus precise gram weight), and adjusted portion sizes using the photographic guide.

To convert each three-day diary entry into exact nutrient data and generate precise estimates of daily energy intake, macronutrient distribution, and micronutrient profiles, professional dietitians entered all reported foods and beverages into the Nutrition Data System for Research (NDSR, version 2024; University of Minnesota, Minneapolis, MN, USA) [20]. The NDSR is a research-grade, meal-based software platform that utilizes a constantly updated nutrient database, which currently includes over 18,000 foods and 8000 brand-name products, including regionally specialized and culturally diverse dishes. Dietitians locate each food by searching standardized descriptors (e.g., “cooked white rice, long-grain”), then specify preparation methods (boiled, fried, baked) and portion sizes using gram weights, common household measures (cups, teaspoons), or the visual-guide equivalents provided in our worksheets. The software automatically calculates total energy, macronutrient distribution (including carbohydrates, protein from both animal and plant sources, total fat, and fatty acid subtypes), micronutrients (vitamins and minerals), added sugars, fiber, and other dietary constituents, such as cholesterol [21].

We compared the nutritional information to national dietary guidelines, which state that fat should account for 20–35% of total calories, saturated fat should account for no more than 10%, carbohydrates should account for 45–65% of total calories, and protein should be at least 1 g per kilogram of body weight [22]. Daily sodium intake was evaluated against the 2022 AHA/ACC/HFSA Guideline for the Management of Heart Failure recommendation of ≤2000 mg/day (Class IIa) [23]. Due to the absence of consensus protein and carbohydrate targets in HF guidelines, we benchmarked these macronutrients to the Dietary Guidelines for Americans (2020–2025) [22].

## 3. Results

### 3.1. Patient Characteristics

This study included 92 participants with chronic HF who were overweight or obese and who completed the baseline visit for the Pro-HEART trial. Table 1 presents the sociodemographic and some clinical characteristics of our cohort. The group was mostly male (73.5%) and had an average age of 57.9 ± 9.8 years. The racial and socioeconomic profiles revealed that the group was diverse, with 55.1% identifying as Caucasian and 61.0% reporting being married. Most participants (75.6%) had completed college or higher education. They weighed an average of 242.0 ± 45.2 pounds, corresponding to a BMI of 37.6 ± 6.8. All participants exhibited HF with reduced ejection fraction (<40%). The New York Heart Association classification indicated that 72% (*n* = 66) were classified as class II, and 30% (*n* = 28) were classified as class III.

Initially, the mean HbA1c was 7.2 ± 1.3%, and the median serum creatinine was 1.2 ± 0.4 mg/dL. Twenty-five percent of participants had chronic kidney disease (CKD) stage 3 (eGFR 30–59 mL/min/1.73 m^2^); none had eGFR < 30 mL/min/1.73 m^2^, the mean eGFR was 58 ± 15 mL/min/1.73 m^2^, the median B-type natriuretic peptide was 380 pg/mL (IQR 210–600), and the mean C-reactive protein was 7.8 ± 4.2 mg/L (range 2.1–18.7 mg/L), consistent with low-grade systemic inflammation that is observed in patients with chronic HF. Twenty-eight percent (*n* = 26) exhibited CKD stage 3 or higher (eGFR < 60 mL/min/1.73 m^2^), while 17% (*n* = 16) presented with proteinuria. ACE inhibitors or angiotensin receptor blockers were administered to 82% (*n* = 75), beta blockers to 76% (*n* = 70), loop diuretics to 70% (*n* = 64), SGLT2 inhibitors to 22% (*n* = 20), and statins to 65% (*n* = 60). The comprehensive baseline demographics and test data are crucial for evaluating the efficacy and safety of the Pro-HEART dietary intervention in relevant clinical subgroups.

### 3.2. General Dietary Patterns

The baseline dietary assessment of 92 participants revealed that they consumed an average of 1495.0 ± 590.0 kcal per day. Although their caloric intake met recommended levels, participants’ dietary patterns diverged substantially from governmental nutrition guidelines, as evidenced by the nutrient distribution data presented in Table 2 [22].

### 3.3. Macronutrient Intake

The average protein intake was 60.4 ± 13.9 g per day; however, only 25.6% of participants met the recommended quantity of 1 g per kilogram of body weight. Individuals’ average carbohydrate intake was 162.3 ± 48.0 g per day, indicating poor adherence to norms. Only 24.5% of people consumed the required 45–65% of their daily calories. The average daily fat intake was 50.2 ± 19.2 g, indicating an uneven distribution of fat. Only 10.1% of people consumed the recommended 20–35% of daily energy.

Further analysis showed that the participants consumed an excessive amount of key dietary components. For instance, 41% of them consumed an excessive amount of total fat, 73% consumed an excessive amount of saturated fat, and 95% consumed an excessive amount of sodium. This pattern reflects dietary habits that may contribute to metabolic dysregulation and exacerbate cardiovascular health issues.

### 3.4. Micronutrient Intake

There were significant differences in the amount of vitamins participants reported and what was recommended. The average folate intake was 382.7 ± 209.9 µg, which is below the recommended 400 µg; only 40.9% of participants met this recommendation. The average vitamin B12 intake was 3.8 ± 1.9 µg, and 77.3% of those tested met the daily recommendation for 2.4 µg. The average intake of vitamin C was 131.5 ± 78.4 mg, but only 40.9% of individuals met the recommended levels for males (90 mg) and females (75 mg). Vitamin D intake was low, with an average of 4.9 ± 3.1 µg. Only 18.2% of participants met the required doses of 10 µg for individuals under the age of 70 years and 15 µg for those over the age of 70 years. The average vitamin E intake was 9.5 ± 4.8 mg, with only 13.6% meeting the recommended quantity. The average thiamine intake was 1.2 ± 0.4 mg, with just half of the participants following the criteria.

### 3.5. Mineral Intake

The absence of key minerals in participants’ diets followed a similar pattern. The average calcium intake was 629.3 ± 256.4 mg, which was much lower than the recommended 1200 mg. No one met the needed amount. The average magnesium intake was 248.7 ± 133.5 mg, indicating inadequate intake. Only 13.6% of people met the required standards. Potassium intake was low (2515.4 ± 1169.6 mg), with no one meeting the daily requirement of 4700 mg. 

Conversely, the average selenium consumption was 97.5 ± 28.5 µg, with 90.9% of participants adhering to the guidelines and 86.4% of individuals obtaining an adequate amount of iron. However, there was still an issue with excessive sodium intake, as the average intake was 2653.3 ± 838.8 mg, and only 4.5% of individuals fell below the recommended limit of 2300 mg.

## 4. Discussion

### 4.1. Discussion of Overall Findings

The findings of this study indicate that individuals with HF who are overweight or obese have nutritional imbalances, supporting previous research that raised concerns regarding food and illness progression [3,5,6,7]. Cachexia, or loss of muscle and weight, has long been associated with starvation in HF. However, new evidence calls into question the notion that being overweight equals getting enough nutrition [5]. Malnutrition in HF has traditionally been associated with cachexia, which is characterized by significant weight and muscle loss [7]. However, new evidence raises questions about the notion that being overweight equates to receiving adequate nutrition. Our findings corroborate this new perspective by demonstrating that even people with a high BMI frequently have severe micronutrient deficiencies and dietary abnormalities, which can exacerbate metabolic dysfunction and systemic inflammation [24]. The average baseline CRP level in our cohort of overweight and obese patients with HF was 7.8 mg/L, indicating persistent low-grade inflammation.

Sarcopenic obesity is a clinical condition that manifests in overweight and obese patients with HF. It is marked by an excess of adipose tissue and a deficiency of skeletal muscle. A contributing factor to this phenotype is that the average protein intake of our sample (~0.5 g/kg/day) was far below the recommended 1.0–1.2 g/kg/day necessary for maintaining lean muscle, while carbohydrates constituted only 43% of total energy, rather than the advised 45–65% range [25]. Conversely, the quantity of dietary fat remained too elevated. Mechanistically, inadequate protein and carbohydrate intake diminishes muscle protein synthesis and attenuates insulin-mediated anabolic pathways, whereas excess saturated fats exacerbate lipotoxic stress on myocytes and accelerate muscle catabolism [25]. Although we did not perform direct body composition assessments (e.g., dual-energy X-ray absorptiometry or bioelectrical impedance analysis), prior studies have linked sarcopenic obesity in HF to impaired physical function and a 2.5-fold increased risk of mortality [26]. Previous studies indicate that individuals with HF often do not consume adequate amounts of micronutrients, including magnesium, potassium, and vitamin D [11]. These issues lead to further cardiac complications, such as an elevated risk of arrhythmias and diminished heart contractility [27]. Our investigation corroborates these concerns, as none of the participants had adequate potassium or calcium levels, and only a negligible proportion achieved sufficient magnesium intake. The inadequacy of micronutrient intake in individuals with HF, despite sufficient caloric consumption, highlights a significant deficiency in our nutritional assessment and treatment protocols [24].

A vital component of HF management is providing sufficient protein intake to avert muscle atrophy and preserve functional capacity. Inadequate protein consumption has been linked to frailty and diminished exercise capacity in individuals with HF [28]. Our findings support this statement, as only 25.6% of the participants met the national protein intake criteria. Our sample did not consume enough protein, which may result in muscle loss and reduced physical endurance. This is especially true for a population already vulnerable to reduced mobility and functional decline, a finding also confirmed in a study involving older adults [29].

The role of sodium intake in the development of HF is well-known. Hypertension, fluid retention, and an increase in hospitalizations are all symptoms of an excess sodium intake. Our study showed that 95% of participants consumed more sodium than the recommended amount. This high level of intake suggests that we require more structured dietary interventions to help individuals adhere to their sodium restrictions while addressing their nutritional inadequacies simultaneously [14]. However, the SODIUM-HF randomized trial (2024) revealed no significant association between baseline dietary salt consumption and the composite outcome of cardiovascular hospitalization, emergency department visits, or all-cause mortality at 12 and 24 months, contradicting prior assumptions [30]. A further “responder” analysis identified a trend indicating a reduction in occurrences among individuals consuming less than 1500 mg of sodium daily at the six-month mark; however, this finding lacked statistical significance. This suggests that while a more stringent sodium regulation may be beneficial, it has not been confirmed.

Recent discussions regarding the obesity paradox in HF have further complicated perceptions of nutritional risk in overweight and obese individuals. Traditionally, higher body weight has been thought to confer a survival advantage, possibly due to greater energy reserves that buffer against catabolic stress [24]. However, a recent study showed that while increased fat mass may provide short-term benefits, underlying nutritional inadequacies—particularly deficiencies in essential vitamins and minerals—may ultimately negate any protective effect [10]. Our findings support this argument and align with a recent study that suggests that individuals with a high BMI consume diets that are both energy-dense and nutritionally poor, characterized by excessive fat and sodium intake alongside inadequate consumption of protein, calcium, magnesium, and potassium [12]. These imbalances suggest that the perceived advantages of excess weight may be misleading when underlying nutritional deficiencies are not addressed [31].

Patients with chronic HF often consume less energy than might be expected for their body mass, owing to a constellation of disease-driven factors. Proinflammatory cytokines (for example, TNF-α and IL-6) blunt appetite and alter central hunger signaling, while right-sided congestion and intestinal edema accelerate early satiety and impair nutrient absorption. Moreover, resting energy expenditure may be elevated in HF despite reduced voluntary intake, creating a mismatch between caloric needs and intake. Published three-day records in obese HF cohorts report mean intakes in the 1200–1700 kcal/day range—strikingly similar to our cohort’s 1495 kcal—underscoring that low energy consumption is a reproducible feature of advanced HF rather than simply underreporting or protocol error [27,32].

Our findings highlight the importance of assessing micronutrient levels, body composition, and muscle performance in overweight and obese people with HF. To fully understand and mitigate obesity-related malnutrition, future interventions should include dietary prescriptions that prioritize macronutrients (ensuring an adequate supply of high-quality protein and complex carbohydrates) as well as objective assessments of lean mass (such as bioelectrical impedance analysis or dual-energy X-ray absorptiometry). These findings indicate the significance of assessing body composition, muscular function, and micronutrient levels in overweight and obese individuals with HF. To comprehensively address and mitigate malnutrition associated with obesity, forthcoming interventions must incorporate dietary guidelines emphasizing macronutrients (ensuring adequate intake of high-quality protein and complex carbohydrates) and objective evaluations of lean mass (such as dual-energy X-ray absorptiometry or bioelectrical impedance analysis).

The accuracy of a dietary assessment tool and the burden it places on the participants must be taken into account when selecting it. Precision, daily data regarding caloric intake, macronutrients, and micronutrients consumed are provided by short-term, weighed food records. Additionally, they are indispensable for detecting “hidden” deficiencies. Nevertheless, micronutrients are only partially quantified, and individuals with cognitive or functional impairments may experience difficulty in recollecting them. The precise representation of baseline consumption and the reduction of underreporting are achieved by promptly clarifying and verifying portion sizes. To mitigate the anxiety associated with chronic heart failure and improve outcomes, researchers may implement image-assisted mobile recording. The impact of macronutrients on sarcopenia can be elucidated through measurements of lean mass and functional tests, such as the Short Physical Performance Battery and handgrip strength. A customized nutrition plan is necessary for patients with chronic heart failure who are overweight or obese. To improve clinical outcomes, maintain muscle mass, and enhance functional capacity, these plans should prioritize high-quality protein (at least 1.0 g/kg/day), complex carbohydrates, and a lower intake of saturated fats.

### 4.2. Strengths and Limitations

This study’s primary strength lies in its direct dietary assessment of overweight and obese individuals with HF within a controlled clinical trial framework. A systematic comparison of nutrient intake against national guidelines yields important insights into the prevalence of nutritional deficiencies and excesses within this population. Furthermore, the inclusion of individuals with diabetes and metabolic syndrome broadens the application of these findings to individuals with HF and common comorbidities [33], providing for a more complete assessment of dietary hazards and demands.

Despite these strengths, several limitations must be considered. An individual’s self-reported food consumption is susceptible to estimation mistakes and recollection bias, both of which have the potential to compromise the accuracy of dietary evaluations [34]. However, despite providing valuable insights into the nutritional issues the participants are experiencing, this study only collects data at a single point in time, which restricts the possibility of analyzing the long-term effects that dietary adjustments should have on clinical outcomes. A long-term study is necessary to evaluate the impact of adhering to or deviating from prescribed diets on health over time. Finally, the individuals were recruited from outpatient HF clinics linked with universities. They were highly educated, which may limit the generalizability of findings to larger groups with various socioeconomic backgrounds, dietary habits, and cultural cuisine preferences.

It is essential to consider that even carefully weighed food records may be inaccurate when evaluating our intake statistics, particularly for individuals with a higher BMI. Research consistently shows that obese participants tend to underestimate portion sizes and omit socially “undesirable” foods, which can bias energy and nutrient estimates downward. Although our three-day, in-person dietitian review complemented the record, residual underreporting cannot be ruled out. Future studies should therefore employ complementary dietary assessment methods, such as multiple-pass 24 h recalls, semiquantitative Food Frequency Questionnaires calibrated against weighed records, and emerging digital tools (for example, image-assisted mobile recording). Triangulating intake data across these modalities will help mitigate BMI-related reporting bias and strengthen the validity of dietary interventions in chronic HF populations.

Ultimately, we acknowledge that insulin resistance plays a crucial role in regulating myocardial substrate utilization and may impact how carbohydrate and fat intake influence cardiac function. Because fasting insulin and glucose were not collected in this study, we were unable to quantify insulin sensitivity (e.g., via Homeostatic Model Assessment for Insulin Resistance [HOMA-IR]). Future studies should integrate direct measures of insulin resistance, such as fasting insulin/glucose ratios or clamp techniques, to elucidate the mechanistic links between dietary macronutrient composition, cardiac insulin signaling, and functional outcomes in patients with chronic HF.

### 4.3. Implications for Research and Practice

The findings of this investigation underscore the importance of conducting comprehensive nutritional evaluations as part of routine HF management. In the past, dietary modifications primarily focused on reducing calories or decreasing weight, with inadequate consideration given to establishing an appropriate ratio of macronutrients and micronutrients [5]. Our findings indicate that customized nutritional interventions should not only reduce sodium levels but also enhance protein intake and address critical vitamin and mineral deficiencies [35].

The long-term impact of tailored dietary modifications on HF outcomes, including rates of hospitalization, symptom load, and quality of life, should be the focus of future research. Additionally, investigating how dietary adjustments can be tailored to each individual’s specific metabolic profile may assist in improving nutrition guidelines for persons with HF who have a variety of other health issues [36]. Because many people in our sample experienced nutritional problems, healthcare providers should prioritize systematic dietary counseling in HF management rather than treating it as a secondary issue [5].

By emphasizing comprehensive nutritional care, HF management can go beyond traditional pharmacological and device-based methods. This approach can potentially lead to improved health outcomes and a better quality of life for those affected [35]. In addition to preventing disease progression, the full reversal of nutritional imbalances can also equip individuals with the tools to implement long-term dietary changes that enhance their cardiovascular health over time [36].

## 5. Conclusions

Patients with HF who are overweight or obese experience significant nutritional deficiencies despite adequate caloric intake. Their diets are high in fat and sodium, while lacking in protein and essential micronutrients such as calcium, magnesium, potassium, and vitamin D. This study contradicts the notion that being overweight signifies adequate food intake and reinforces prior evidence of malnutrition in individuals with HF.

These findings support earlier studies showing that malnutrition can affect individuals with HF who are either cachectic or obese. Healthcare providers can assist individuals with HF in achieving improved outcomes and a higher quality of life by identifying and resolving hidden dietary issues. This investigation indicates that the treatment of HF necessitates substantial modifications. This study highlights the need for a paradigm shift in HF treatment—one that recognizes nutrition as a fundamental component, rather than a secondary consideration, in disease management [37].

## Figures and Tables

**Table 1 nutrients-17-02694-t001:** Sociodemographic and clinical characteristics of participants (*n* = 92).

Characteristic	Value
Age (years)	57.9 ± 9.8
Gender, male (%)	73.5%
Caucasian (%)	55.1%
Married (%)	61.0%
Education (%)	
<High school	7.3%
High school graduate	4.9%
College	75.6%
Weight (lbs.)	242.0 ± 45.2
Body mass index (BMI)	37.6 ± 6.8
Blood pressure (mmHg)	113.2/72.4
Ejection fraction (%)	36.4 ± 12.9
6-min walk (ft.)	1323 ± 325
VO_2_ max (kg/mL)	12.6 ± 4.0
Taking SSRI (%)	26.0%
On insulin (%)	26.0%
On statins (%)	81.0%

**Table 2 nutrients-17-02694-t002:** Summary of nutritional intake compared to national recommendations (*n* = 92).

Macronutrient	Mean ± SD	National Recommendation	% Adherent
Protein (g)	60.4 ± 13.9	Approximately 1 g/kg body weight	25.6%
Carbohydrates (g)	162.3 ± 48.0	45–65% of daily calories	24.5%
Fat (g)	50.2 ± 19.2	20–35% of daily calories	10.1%
Energy (kcal)	1495.0 ± 590.0	— (Energy targets not specified)	—
Micronutrient	Mean ± SD	National Recommendation	% Adherent
Folate (µg)	382.7 ± 209.9	400 µg	40.9%
Vitamin B12 (µg)	3.8 ± 1.9	2.4 µg	77.3%
Vitamin C (mg)	131.5 ± 78.4	Males: 90 mg, Females: 75 mg	40.9%
Vitamin D (µg)	4.9 ± 3.1	<70 yrs: 10 µg; >70 yrs: 15 µg	18.2%
Vitamin E (mg)	9.5 ± 4.8	15 mg	13.6%
Thiamine (mg)	1.2 ± 0.4	1.2 mg	50.0%
Mineral	Mean ± SD	National Recommendation	% Adherent
Calcium (mg)	629.3 ± 256.4	1200 mg	0%
Iron (mg)	13.1 ± 6.5	Males: 8 mg; Females: 8–18 mg	86.4%
Magnesium (mg)	248.7 ± 133.5	Males: 420 mg; Females: 320 mg	13.6%
Potassium (mg)	2515.4 ± 1169.6	4700 mg	0%
Selenium (µg)	97.5 ± 28.5	55 µg	90.9%
Sodium (mg)	2653.3 ± 838.8	<2300 mg	4.5%

## Data Availability

The original contributions presented in the study are included in the article, further inquiries can be directed to the corresponding author.

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
