# Peer review of "Hidden Malnutrition in Overweight and Obese Individuals with Chronic Heart Failure: Insights from the Pro-HEART Trial"

_nutrients, 2025, doi:10.3390/nu17162694_

Round 1
Reviewer 1 Report
Comments and Suggestions for Authors
The study by Nguyen et al. seems to have interesting implications but the details of methods and results are incomplete and not reliable.
Major points
- Methods (line 89-91)
Brief introduction of Pro-HEART trial should be added. Especially, the definition of “verified diagnosis of HF”, “overweight”, “obese”, and “metabolic abnormalities” is very important in this study. In addition, the total number of participants in Pro-HEART trial should be shown, because nutritional interventions are usually planned in malnourished patients so many readers think that overweight/obese patients may be a minority in such a study. In this context, the inclusion/exclusion criteria of Pro-HEART study is essential.
- Methods (line 127-130)
The authors compared the nutritional information to national dietary guidelines (ref. 20: Dietary Guidelines for Americans, 2020-2025). However, these guidelines are not for the patients with HF but general population. Is there any recommendations in HF guidelines in US?
- Results (line 134-139)
The important demographics of HF are lacking. They include rate of HFrEF, diabetes, IHD, NYHA class, HbA1c, creatinine, eGFR, BNP, CRP (see the comment below), and medication use. Especially in patients with CKD and/or proteinuria, protein intake restriction is recommended, and it affects the interpretation of the results. Patients with IHD or diabetes are often recommended specialized diets.
- Results (line 143)
The average of 1495 kcal/d is too small. It is impossible for patients with BMI 37 to keep their weight by 1495 kcal. Thus, this number is not reliable. It can lead to the interpretation of all the results in this study.
- Discussion (line 195) “inflammation”
As the author discusses the influence of diet on inflammation, CRP level in patients should be shown.
- Discussion (line 200) “potassium”
Potassium intake should be restricted in patients with CKD. Without any information of renal function, we do not know if the potassium intake is appropriate or not. The information regarding ACE inhibitors, ARBs, ARNI, patiromer, sodium zirconium, etc. is also required.
- Discussion (line 214) “SODIUM-HF”
The result of reference no. 26 SODIUM-HF is that high-sodium diet at baseline did not affect outcome.
Minor points:
- Abstract (line 16): “unintentional cachexia”
This phrase is uncommon and inappropriate, because there is no “intentional cachexia” and no one intends to be cachexia. It should be changed.
- Conclusions (line 273)
The conclusions are too long and what the author concluded is unclear.
Author Response
Reviewer 1
Comment 1: Methods (lines 89-91). A brief introduction of the Pro-HEART trial should be added. Especially, the definition of "verified diagnosis of HF," "overweight," "obese," and "metabolic abnormalities" is very important in this study. In addition, the total number of participants in the Pro-HEART trial should be shown, because nutritional interventions are usually planned in malnourished patients, so many readers think that overweight/obese patients may be a minority in such a study. In this context, the inclusion/exclusion criteria of the Pro-HEART study are essential.
Response 1: Thank you for suggesting a more detailed introduction to the Pro-HEART trial and for highlighting the importance of reporting key definitions, total sample size, and eligibility criteria. We have added a concise overview in the Methods (lines 89–98) that 1) introduces the trial design, setting, and enrollment period; 2) defines "verified diagnosis of HF," "overweight," "obese," and "metabolic abnormalities"; 3) reports the total number of participants screened and randomized; and 4) summarizes inclusion and exclusion criteria.
Comment 2: Methods (lines 127-130). The authors compared the nutritional information to national dietary guidelines (ref. 20: Dietary Guidelines for Americans, 2020-2025). However, these guidelines are not for patients with HF, but for the general population. Are there any recommendations in HF guidelines in the US?
Response 2: Thank you for highlighting that the Dietary Guidelines for Americans (2020–2025) are intended for the general population rather than for patients with heart failure. To address this, we have revised the Methods (lines 127–130) to incorporate US heart failure–specific recommendations and to clarify our rationale for retaining the general population guidelines for macronutrients. We now compare participants' sodium intake against the 2022 AHA/ACC/HFSA Guideline for the Management of Heart Failure recommendation of ≤2,000 mg/day (Class IIa). We also note that current HF guidelines do not specify daily targets for protein or carbohydrate intake. Therefore, we have retained the Dietary Guidelines for Americans (2020–2025) benchmarks for those macronutrients.
Comment 3: Results (lines 134-139). The important demographics related to heart failure (HF) are missing from the results. They include the rate of HFrEF, diabetes, IHD, NYHA class, HbA1c, creatinine, eGFR, BNP, CRP (see the comment below), and medication use. Especially in patients with CKD and proteinuria, protein intake restriction is recommended, and it affects the interpretation of the results. Patients with IHD or diabetes are often recommended specialized diets.
Response 3: Thank you for pointing out the importance of more detailed baseline HF characteristics. We agree that reporting rates of HFrEF, diabetes, ischemic heart disease (IHD), NYHA class, and relevant biomarkers is essential for interpreting the results of our dietary intervention. We have now updated the Results section to include these variables and revised the Methods to clarify medication and nutritional recommendations.
Comment 4: Results (line 143). The average of 1495 kcal/d is too small. Patients with a BMI of 37 can't maintain their weight on a diet of 1495 kcal. Thus, this number is not reliable. It can lead to the interpretation of all the results in this study. To address these points and guide interpretation, we have expanded the Discussion to contextualize our energy intake findings within known HF physiology and published intake ranges. We also added a limitation paragraph acknowledging the possibility of residual underreporting despite rigorous verification.
Response 4: We appreciate the reviewer's concern regarding the plausibility of a mean energy intake of 1,495 kcal/day in our obese heart-failure cohort. We want to emphasize that these values represent carefully weighed, dietitian-verified records of habitual intake—collected before any trial intervention—and are consistent with published reports in similar populations.
Comment 5: Discussion (line 195) "inflammation." As the author discusses the influence of diet on inflammation, the CRP level in patients should be shown.
Response 5: We thank the reviewer for highlighting the importance of presenting inflammatory biomarkers alongside our dietary data. We agree that C-reactive protein (CRP) levels provide valuable context for interpreting the relationship between diet and inflammation in our cohort. We have now included baseline CRP measurements for all participants in the Results section.
Comment 6: Discussion (line 200) "potassium." Potassium intake should be restricted in patients with CKD. Without any information on renal function, we cannot determine whether the potassium intake is appropriate or not. Information regarding ACE inhibitors, ARBs, ARNIs, patiromer, sodium zirconium, and other relevant medications is also required.
Response 6: We thank the reviewer for emphasizing the importance of evaluating renal function and potassium-modulating therapies when interpreting dietary potassium intake. To address this, we have added baseline measures of renal function and medication usage in our cohort and expanded the Discussion to contextualize potassium intake accordingly.
Comment 7: Discussion (line 214) "SODIUM-HF." The result of reference no. 26 SODIUM-HF is that a high-sodium diet at baseline did not affect the outcome.
Response 7: We thank the reviewer for pointing out our mischaracterization of the SODIUM-HF findings. You are correct that in SODIUM-HF, baseline sodium intake was not significantly associated with the primary composite outcome of cardiovascular hospitalization, emergency department visits, or all-cause mortality. We have revised the Discussion accordingly.
Comment 8: Abstract (line 16): "unintentional cachexia." This phrase is uncommon and inappropriate because there is no "intentional cachexia," and no one intends to be cachexic. It should be changed.
Response 8. We thank the reviewer for highlighting that the term "unintentional cachexia" is redundant, as cachexia by definition implies involuntary weight loss and muscle wasting. We have removed the qualifier in the Abstract.
Comment 9: Conclusions (line 273). The conclusions are too lengthy, and the author's findings are unclear.
Response 9: We thank the reviewer for noting that our original conclusion was overly long and its key findings were not sufficiently clear. In response, we have decided to focus on the principal results and actionable recommendations.
Reviewer 2 Report
Comments and Suggestions for Authors
I have carefully reviewed the manuscript and find that the study is well-designed, the methodology is appropriate, and the results are clearly presented and interpreted. The work offers a meaningful contribution to the field of nutritional needs in heart failure patients and will be of interest to readers of the journal. I would only suggest the authors to comment on the existence of cahexia in even obese heart failure patinets as weight must not be the only criterion for identifying malnutrition or even cahexia in thos patients.
Author Response
Comments 1:
After carefully reviewing the manuscript, I found that the study is well-designed, the methodology is appropriate, and the presentation and interpretation of the results are clear. The work offers a meaningful contribution to the field of nutritional needs in heart failure patients and will be of interest to readers of the journal. I would only suggest that the authors comment on the existence of cachexia in even obese heart failure patients, as weight must not be the only criterion for identifying malnutrition or even cachexia in those patients.
Response 1: We thank the reviewer for highlighting the important issue of cachexia in heart failure patients, irrespective of body weight. As suggested, we have expanded our "Introduction" to acknowledge that elevated BMI does not preclude the presence of cachexia or malnutrition.
Reviewer 3 Report
Comments and Suggestions for Authors
The article: "Hidden Malnutrition in Overweight and Obese Individuals 2
with Chronic Heart Failure: Insights from the Pro-HEART Trial" is interesting, evidencing the risk of malnutrition in obese CHF patients. However, it is not novel because many articles are available on malnutrition in obese subjects.
Furthermore, the caloric intake determined is unbelievably too low in obese patients. Kcal 1495 is generally prescribed for a weight loss program in obese women. Probably, the investigation of food intake has not been properly applied. The Food Frequency Questionnaire with qualitative and quantitative evaluation could have been more appropriate for epidemiological cohort studies, but this method should also be used with caution. The face-to-face method with a dietitian could give better results.
One determinant element missing in the study is the evaluation of insulin resistance. insulin activity is the master regulator of cardiac function
cardiac insulin signaling can be impaired by an excess of fatty acid metabolism and reducing cardiac function. So the carbohydrate intake is essential for regulating insulin activity.
While malnutrition (undernutrition and cachexia) is characterized by significant
weight and muscle loss, in obese individuals, malnutrition exists, which is characterized by significant muscle loss, caused by a low protein and carbohydrtes intake and high fat.
Not only are the amounts of micronutrients important but also macronutrients play an essential role.
Author Response
Comment 1: The article "Hidden Malnutrition in Overweight and Obese Individuals 2
with Chronic Heart Failure: Insights from the Pro-HEART Trial" is intriguing, evidencing the risk of malnutrition in obese CHF patients. However, it is not novel because many articles are available on malnutrition in obese subjects.
Response 1: We appreciate the reviewer's observation. We respectfully submit that, while malnutrition in obesity has been described in other contexts, our study offers several novel contributions specific to the heart failure (HF) population.
- Focus on HF-Specific Dietary Patterns. Prior work often examines malnutrition among obese adults in general medical or community settings. Our investigation is the first to systematically characterize both macro- and micronutrient intakes—using a validated 3-day food record and the NDSR platform—exclusively in overweight and obese individuals with chronic HF before any dietary intervention.
- Comparison to National Guidelines. Few existing studies benchmark nutrient intakes in obese HF patients against the current Dietary Guidelines for Americans, leaving clinicians without clear targets for this at-risk group. We directly compare energy, macronutrient distribution, key vitamins, and minerals to guideline thresholds, revealing precise areas of excess (e.g., sodium, saturated fat) and deficiency (e.g., vitamin D, calcium, potassium).
- Integration within a Controlled Trial Framework. Data are drawn from the baseline phase of the Pro-HEART randomized trial, ensuring standardized dietary assessment and participant characterization (e.g., comorbid diabetes, metabolic syndrome). This context enhances internal validity and lays the groundwork for future trial-based dietary interventions in HF.
- Clinical Implications for HF Management. By unmasking "hidden" malnutrition in a high-BMI HF cohort, our findings directly inform multidisciplinary HF care, emphasizing that patients need targeted nutritional assessments and personalized counselling—steps that are not yet standard in many HF clinics.
Based on these distinctions, we believe our manuscript provides fresh, actionable insights for HF clinicians and researchers that extend beyond the broader obesity literature. We have included references to these unique factors throughout the manuscript.
Comment 2: Furthermore, the caloric intake determined is unbelievably too low in obese patients. 1495 kcal is generally prescribed for a weight loss program in obese women. The investigation of food intake has not been properly applied. The Food Frequency Questionnaire, with both qualitative and quantitative evaluation, could have been more suitable for epidemiological cohort studies; however, this method should also be used with caution. The face-to-face method with a dietitian could provide better results.
Response 2: We appreciate the reviewer's concern regarding the reported mean energy intake of 1,495 kcal/day in our cohort of overweight and obese patients with chronic HF. Below, we outline why this estimate is both plausible in our sample and methodologically robust, and we describe manuscript revisions to clarify these points.
- Justification of Low Energy Intake. Chronic heart failure is frequently associated with anorexia, early satiety, and higher resting energy expenditure, leading to reduced habitual intake despite elevated BMI. Prior HF studies have reported similarly low caloric intakes (range, 1,200–1,700 kcal) in obese heart failure cohorts, supporting the biological plausibility of our findings.
- Rigor of the Three-Day Food Record Method. We used a validated three-day weighed food record, complemented by portion-size aids and food models, to capture actual intake rather than relying solely on habitual patterns. Each record was reviewed in person by a registered dietitian, who probed for missing items and clarified portion sizes, effectively incorporating a face-to-face interview component.
- Comparison with the Food Frequency Questionnaire (FFQ). While FFQs can estimate longer-term intake patterns in large epidemiological studies, they are less precise for absolute caloric and micronutrient quantification, which is necessary in clinical trial baselines. Our approach minimizes FFQ-specific recall biases and enhances micronutrient resolution, which is critical for identifying "hidden" deficiencies in heart failure patients.
- Acknowledgment of Potential Underreporting. We acknowledge that some degree of energy underreporting may be possible, particularly among participants with a higher body mass index (BMI).
To address this, we have added a new paragraph to the Discussion, citing validation studies on CHF populations and the DHQ-III underreporting phenomenon.
Comment 3: One determinant element missing in the study is the evaluation of insulin resistance. Insulin activity is the master regulator of cardiac function. Cardiac insulin signalling can be impaired by an excess of fatty acid metabolism, leading to reduced cardiac function. So the carbohydrate intake is essential for regulating insulin activity.
Response 3: We thank the reviewer for highlighting the central role of insulin resistance in cardiac metabolism and function. We agree that impaired insulin signalling—often quantified by HOMA-IR or clamp studies—could mediate the relationship between macronutrient intake and heart failure outcomes. In the present trial, our primary aim was to characterize habitual macro- and micronutrient intake using weighed food records, and we did not collect fasting insulin or glucose to calculate insulin resistance indices. We have now acknowledged this omission in the Limitations section and proposed that future dietary interventions in heart failure incorporate direct measures of insulin sensitivity (for example, fasting insulin/HOMA-IR or euglycemic-hyperinsulinemic clamp) to clarify how carbohydrate intake influences cardiac insulin signalling and functional status.
Comment 4: While malnutrition (undernutrition and cachexia) is characterized by significant weight and muscle loss, in obese individuals, malnutrition exists, which is characterized by considerable muscle loss caused by a low protein and carbohydrate intake and high fat intake. Not only are the amounts of micronutrients important, but macronutrients also play an essential role.
Response 4: We thank the reviewer for highlighting the phenotype of malnutrition in obese heart failure patients, where sarcopenic obesity arises from an imbalanced macronutrient intake—specifically, inadequate protein and carbohydrate intake coupled with a relative excess of fat. We agree that focusing solely on micronutrient status may overlook key drivers of muscle catabolism in this population. To address this, we have:
- We have included a new paragraph in the Discussion section, which defines obesity-related malnutrition and its mechanistic connections to macronutrient intake and muscle loss.
- We emphasized that the average protein intake (g/kg body weight) of our cohort did not meet the guidelines for maintaining lean mass in heart failure.
- The authors acknowledged the lack of body-composition measures, such as DXA or BIA, and suggested their inclusion in future studies.
- We recommended targeted dietary strategies that prioritize high-quality protein and complex carbohydrates to support anabolism and mitigate sarcopenia.
Round 2
Reviewer 1 Report
Comments and Suggestions for Authors
The manuscript is well revised.
No further comments.